# Bridging the Capacity Building Gap for Antimicrobial Stewardship Implementation: Evidence from Virtual Communities of Practice in Kenya, Ghana, and Malawi

**DOI:** 10.3390/antibiotics14080794

**Published:** 2025-08-04

**Authors:** Ana C. Barbosa de Lima, Kwame Ohene Buabeng, Mavis Sakyi, Hope Michael Chadwala, Nicole Devereaux, Collins Mitambo, Christine Mugo-Sitati, Jennifer Njuhigu, Gunturu Revathi, Emmanuel Tanui, Jutta Lehmer, Jorge Mera, Amy V. Groom

**Affiliations:** 1ECHO Institute, University of New Mexico Health Sciences Center, 1650 University Blvd. NE, Albuquerque, NM 87102, USA; ndevereaux@salud.unm.edu (N.D.); jlehmer@salud.unm.edu (J.L.);; 2Department of Pharmacy Practice, School of Pharmacy, University of Health and Allied Sciences, Ho PMB 31, Ghana; kobuabeng@uhas.edu.gh; 3Department of Pharmacy Practice, Faculty of Pharmacy and Pharmaceutical Sciences, College of Health Sciences, Kwame Nkrumah University of Science and Technology (KNUST), Kumasi 00233, Ghana; 4Technical Coordination Directorate, Ministry of Health, Accra P.O. Box M-44, Ghana; ewuradwoasakyi29@gmail.gh; 5Doctoral Programme, Interfaculty Initiative in Planetary Health, Nagasaki University, Nagasaki 852-8131, Japan; 6Ministry of Health, Lilongwe 3 P.O. Box 30377, Malawi; hopemchadwala@gmail.com (H.M.C.);; 7ECHO Africa, University of New Mexico Health Sciences Center, Nairobi 00200, Kenya; cmugositati@salud.unm.edu; 8Ministry of Health, Nairobi P.O. Box 30016, Kenya; 9Department of Pathology, Aga Khan University, Nairobi P.O. Box 30270-00100, Kenya

**Keywords:** AMS, AMR, IPC, community of practice, virtual education, e-learning, healthcare facility assessment

## Abstract

Background/Objectives: Strengthening antimicrobial stewardship (AMS) programs is an invaluable intervention in the ongoing efforts to contain the threat of antimicrobial resistance (AMR), particularly in low-resource settings. This study evaluates the impact of the Telementoring, Education, and Advocacy Collaboration initiative for Health through Antimicrobial Stewardship (TEACH AMS), which uses the virtual Extension for Community Healthcare Outcomes (ECHO) learning model to enhance AMS capacity in Kenya, Ghana, and Malawi. Methods: A mixed-methods approach was used, which included attendance data collection, facility-level assessments, post-session and follow-up surveys, as well as focus group discussions. Results: Between September 2023 and February 2025, 77 virtual learning sessions were conducted, engaging 2445 unique participants from hospital-based AMS committees and health professionals across the three countries. Participants reported significant knowledge gain, and data showed facility improvements in two core AMS areas, including the implementation of multidisciplinary ward-based interventions/communications and enhanced monitoring of antibiotic resistance patterns. Along those lines, participants reported that the program assisted them in improving prescribing and culture-based treatments, and also evidence-informed antibiotic selection. The evidence of implementing ward-based interventions was further stressed in focus group discussions, as well as other strengthened practices like point-prevalence surveys, and development or revision of stewardship policies. Substantial improvements in microbiology services were also shared by participants, particularly in Malawi. Other practices mentioned were strengthened multidisciplinary communication, infection prevention efforts, and education of patients and the community. Conclusions: Our findings suggest that a virtual case-based learning educational intervention, providing structured and tailored AMS capacity building, can drive behavior change and strengthen healthcare systems in low resource settings. Future efforts should aim to scale up the engagements and sustain improvements to further strengthen AMS capacity.

## 1. Introduction

Strengthening antimicrobial stewardship (AMS) programs is a widely recognized approach to address the ongoing challenge of antimicrobial resistance (AMR) [1]. AMR poses critical health threats to populations across the globe and impacts all countries and regions [2]. Low- and middle-income countries (LMICs) are disproportionately impacted by AMR, with Sub-Saharan Africa experiencing the largest burden due to a high prevalence of infectious diseases, weak regulatory structures to control antimicrobial quality and access, limited health and laboratory infrastructure, inequitable access to health care, and often inadequate water, sanitation, and hygiene [3,4,5].

Optimizing antimicrobial use in hospitals is crucial for mitigating AMR, yet implementing AMS programs is challenging due to limited resources, varying cultural practices, and contexts [6]. Although AMS interventions initially started in high-income countries, the WHO AMS toolkit, published in 2019, aimed to promote sustainable behavior change in antibiotic prescribing among healthcare facilities in LMICs [7]. Stakeholders, such as the East, Central and Southern Africa Health Community (ECSA), have developed additional tools and guidance for AMS implementation [8]. African countries, including Malawi, Ghana, and Kenya, have incorporated AMS programs into their National AMR Action Plans, establishing dedicated committees within their health governance structures.

Nevertheless, a recent study in selected Kenyan counties highlighted significant gaps in AMS program implementation, including limited access to laboratory services, poor monitoring of treatment adherence, and inadequate policy execution [9]. For Ghana and Malawi, evidence from the literature suggests implementation levels are lower compared to Kenya. For example, an assessment of 15 hospitals in Ghana found sub-optimal performance in almost all AMS program implementation core elements [10] and the 2023 National Program Audit in Malawi showed critical gaps in five out of six hospitals assessed, including problems with leadership commitment, infrequent implementation of critical strategies such as ward rounds, and inconsistent education on AMS for healthcare staff [11].

A key component of AMS programs is education and training of multi-disciplinary health care personnel on AMR, interventions to promote appropriate diagnostics, and prudent prescribing and use of antimicrobials in infection management. Gaps in global AMR knowledge in the human health sector have been documented, and international collaborations have supported in-person training and digital learning in LMICs [12,13,14]. However, many countries do not provide comprehensive AMS training for multi-disciplinary health care providers, and education and training gaps remain [15,16]. In the WHO Africa region, a 2023 review noted that, across several studies, less than 60% of healthcare providers reported good awareness of AMR, and more than half of the countries surveyed (17/36) have little or no continuing professional education in AMR or AMS [17].

Studies on the success of virtual approaches for healthcare worker training on AMR and AMS are limited, though those that we identified note the feasibility and acceptability of virtual approaches. The US CDC and WHO have self-paced online training modules [14,18], and studies in the US found that the use of online e-learning modules for frontline nurses significantly improved their understanding of AMS roles and enhanced confidence in participating in activities [19,20]. A study in Brazil found that a web-based virtual approach with both asynchronous and synchronous activities effectively increased AMS knowledge among medical students in LMICs [21]. Another study, in Ecuador, found both e-learning and face-to-face AMR training to be effective learning strategies, with providers appreciating the flexibility provided by the e-learning approach [22]. Health workers from several countries in Africa reported knowledge gain by using an AMS game designed to support education on AMS [12], and massive open online courses (MOOCs) have also been successful in engaging participants worldwide, addressing AMS learning needs as a complementary tool [23]. Additionally, in Ethiopia and Kenya, a virtual telementoring AMR program for laboratory technicians led to positive changes in laboratory practices [24]. During the COVID-19 pandemic, virtual education and training approaches were widely implemented out of necessity. They offered greater flexibility and reach, and helped address issues such as staff turnover by providing ongoing training opportunities [25,26,27]. In the context of diminishing resources to support in-person trainings and workshops, it is critical to continue strengthening virtual training approaches to build healthcare worker capacity. However, their success depends on being tailored to the cultural and health infrastructure context. They also require that participants have stable internet connectivity [15,28,29,30].

The Extension for Community Health Care Outcomes (ECHO) model is an effective, low-cost approach for ongoing professional development and mentoring through virtual case-based learning. It emphasizes ‘problem-posing education,’ where learners tackle real-world problems rather than relying on memorization [31]. Established in New Mexico in 2003, the ECHO model began to improve the management of underserved hepatitis C patients, and showed that patient outcomes from ECHO-trained providers were comparable to those of specialists [32]. Since its inception, ECHO has expanded globally, with nearly 1000 partners across more than 70 countries and over 8000 programs in diverse healthcare topics [33,34,35,36]. Utilizing video conferencing to convene health care workers for low-dose/high-frequency learning, sessions last from 1 to 1.5 h. Each session includes a brief lecture and case discussions, thus enhancing access to professional development and promoting local expertise.

The ECHO learning model is integral to the Telementoring, Education and Advocacy Collaboration for Health Through Antimicrobial Stewardship (TEACH-AMS) initiative. The initiative launched in January of 2023 as a collaborative effort between the University of New Mexico Health Sciences Center ECHO Institute, Ministries of Health (MoHs), and other stakeholders. This study focuses on the reach and diversity of professionals participating in TEACH AMS programs in Kenya, Ghana, and Malawi, as well as on identifying AMS knowledge increase, implementation of core AMS elements at their healthcare facilities, best practices uptake, and challenges participants experienced in apply knowledge acquired in sessions. Across all three programs, our findings showcase the impact of the TEACH AMS ECHO model in supporting AMS practice and systems changes. Our study sheds light on the mechanisms through which telementoring capacity-building approaches, such as the ECHO learning model, contribute to the implementation of evidence-based knowledge by health care workers in their facilities to combat antimicrobial resistance.

## 2. Materials and Methods

### 2.1. Study Setting

The TEACH AMS ECHO programs were led by AMR focal points in the respective countries’ MoHs and targeted the AMS committees at selected health facilities. However, programs were also open to all interested participants. The TEACH AMS Kenya program was the first to launch in September 2023, followed by Ghana and Malawi in November 2023. A standard curriculum, with 32 modules, was designed in collaboration with the American Society for Microbiology and Africa-based consultants. The curriculum contained AMS-related topics such as the burden of AMR, relevance and need for AMS and IPC, standard treatment guidelines for common infectious diseases, and appropriate antimicrobial use. Additionally, it addressed core principles for designing and implementing AMS programs, hospital-based AMS interventions, and the role of microbiology and diagnostics in addressing AMR. The modules were available to all country TEACH AMS programs (see Appendix A) to be tailored and adapted as needed and used as session didactic presentations. All three countries held bi-weekly ECHO sessions, which included didactic presentations by local and regional SMEs, followed by detailed AMS case presentations by participants, and discussion. Case presentations were often prepared by a multi-disciplinary group of healthcare professionals from the presenting health facility. The programs used a secure web-based data system, the iECHO platform, to manage programmatic and evaluation data collection. Participants join sessions through the iECHO platform and can access all materials presented and discussed during those sessions.

The Kenya TEACH AMS program evolved from an initial AMR laboratory program started in 2018 [24], expanding to the Integrated AMR, AMS, and IPC ECHO to include a broader range of healthcare professionals involved in patient care. The current program is coordinated and led by the Patient Safety Unit (AMR/IPC/PS) within the Kenya MoH. The TEACH AMS program in Ghana officially launched on 3 November 2023, following a month of intensive preparatory training and engagement with key stakeholders. Currently, a dedicated TEACH AMS hub team based at the MoH coordinates and leads the program. Finally, the Malawi TEACH AMS program was launched on 8 November 2023. In its first year, the focus of the program was on building the AMS committee’s capacity to operationalize stewardship programs effectively at different health facilities. As the project progressed, additional TEACH AMS curriculum topics were introduced based on identified gaps and training needs. The Malawi TEACH AMS program is coordinated and led by the Antimicrobial Resistance National Coordinating Centre (AMRNCC) within the Malawi MoH. As part of the initiative, the ECHO Institute conducted in-person training sessions for partner organizations to equip country teams with skills for implementing the model. Additionally, regular collaborative meetings are held and are open to the TEACH AMS organizing teams in each country. These meetings provide a forum for teams to collaborate on troubleshooting challenges, share strategies for engaging diverse audiences, and exchange resources across countries.

### 2.2. Study Design

This study was designed to (i) monitor program reach and participants’ satisfaction, (ii) assess implementation of the WHO core AMS elements in health facilities before and after participation in the program, (iii) describe the implementation of practice and systems changes reported by participants, and (iv) identify barriers to best practice implementation. We applied a mixed-methods design to obtain data including (i) iECHO participants’ attendance and demographic data, (ii) baseline and follow-up assessment of the implementation of AMS program elements in selected facilities, (iii) iECHO post-session satisfaction survey with retrospective pre/post knowledge self-assessment, (iv) follow-up surveys to assess application of knowledge gained, and (v) focus group discussions (FGDs). We did not collect clinical outcomes data, as they were outside the scope of this study and would have required logistical and financial resources that were unavailable to the study team.

We conducted online baseline and follow-up facility assessments with selected participating sites to understand the extent to which facilities could implement the AMS program core elements before and after participating in the TEACH AMS programs. Participants’ sites were selected by in-country programs teams and had to have a minimum of training and infrastructure to implement the necessary AMS actions at the healthcare facility. Therefore, most of the facilities selected had received funding or participated in other AMS trainings prior to joining the TEACH AMS initiative. Additional criteria included representation of different types of hospitals and geographic distribution in the countries. We used REDCap (Research Electronic Data Capture) to develop a virtual assessment tool with input from stakeholders working in the field of AMS prior to the launch of the programs [37]. The tool contains all healthcare facility core elements for AMS programs outlined in the WHO Practical Toolkit checklist, organized into three levels, from basic (primary) to advanced (tertiary) [7]. Local teams in each country coordinated data collection between August and October 2023 (baseline) and September and December 2024 (follow-up).

We collected feedback after each session using an online survey via iECHO, a standard evaluation tool available to all international ECHO programs. This survey, adapted from the CDC’s evaluation framework [38], assesses session satisfaction, relevance to participants’ work, and changes in knowledge. To assess self-reported knowledge change, we used a retrospective pre- and post-design to avoid response shift bias [39], since participants tend to overestimate their knowledge level before learning sessions. The post-session survey has been successfully used in various international contexts for over three years, providing valuable feedback for program improvement and demonstrating its adaptability in gathering meaningful insights [27].

After the first six sessions, we implemented a follow-up survey in iECHO, informed by attendance and post-session survey data. The survey included both open-ended and multiple-choice questions. The main goal of the follow-up survey was to identify changes participants reported implementing as a result of what they learned in the sessions, either in their individual practice or in systems at their facility. Subsequently, ECHO evaluators conducted three FGDs, one for each program, to gather more nuanced information on practice changes, facility changes, and any perceived barriers or needs experienced by participants. FGD invitations were sent to any participants who responded positively to a question included in the follow-up survey about their willingness to participate in a FGD. We were aware of the high commitment that a FGD represents for participants who are busy health care workers, and there were no incentives for participation.

### 2.3. Study Analysis

The findings presented in this study include data collected between July 2023 and February 2025. We used descriptive statistical analysis to investigate program reach, profession distribution, session satisfaction levels, and practice or systems changes in follow-up surveys’ closed questions. We employed the paired Wilcoxon signed rank sum test to analyze paired data for before- and after-session knowledge levels reported in post-session surveys and, for the facility assessment data, to examine differences in the implementation stages (implemented, partial, or not implemented) of AMS elements for each facility baseline and follow-up data pair. Given the small sample size and the presence of ties in the facility assessment data, Pratt’s modification was used to calculate *p*-values. The analysis was conducted using R [40], RStudio v 2024.12.1+563 [41], the MASS package [42], and the coin package [43].

FGDs were recorded, transcribed, and systematically analyzed using conceptual content analysis for the identification of the presence and frequency of key explicit and latent themes [44]. Evaluators met with AMS SMEs in each country to refine definitions of the coding scheme, particularly those requiring specific expertise and understanding of AMS practices and strategies. NVivo 14 software was used for qualitative analysis of FGD transcripts [45]. Qualitative data from open-ended questions in follow-up surveys were analyzed using the Microsoft Excel spreadsheet editor.

## 3. Results

### 3.1. Attendance Reach

Between September 2023 and February 2025, the Kenya, Ghana, and Malawi TEACH AMS programs held 77 learning sessions with 2445 unique participants. While most participants were from the program host countries, about 8% of participants were from 52 other countries. The average number of sessions attended by participants was four, with 62% attending two or more sessions and 9% attending 10 or more (Table A1). Out of the 2445 unique participants, 1977 attended the Kenya program, a considerably larger participation compared to 290 in the Ghana program and 208 in the Malawi program (30 participants attended more than one country’s program). While participants from diverse professions joined all programs, including pharmacists, laboratory personnel, physicians, and nurses, pharmacists were the most predominant for the Kenya program (61.3% of participants). Ghana’s and Malawi’s most extensive participation was of physicians or physician assistants (36.6% and 37.9%, respectively), with other categories more evenly distributed than in the Kenya program (Figure 1).

### 3.2. Session Satisfaction and Knowledge Gain

We received post-session responses for 68 of the 77 sessions across all three programs. Sessions in which respondents did not complete the survey were due to issues with users not being familiar with the process and a lack of time for program teams to share the survey link during the session. The post-session response rate for those 68 sessions was 8.2% (725 out of 8873 attendances), ranging from 7% and 7.5% for Kenya and Malawi, respectively, and 15% for Ghana. Respondents were highly satisfied with the sessions: 93.8% reported the sessions were well-balanced between lecture and interactivity, 93% rated the sessions as either extremely or very relevant to their work, 77.2% stated that they would definitely recommend the session to a colleague, and 86.1% indicated they would definitely use what they learned in their work. Respondents were asked to rate their knowledge of the session topics, from “Not at all knowledgeable” to “Extremely knowledgeable”, before and after participation. There was a significant increase in knowledge of the session topic (V = 7061.5, *p* < 0.0001), with 70.8% of respondents reporting an increase in their knowledge after session participation (Figure 2).

### 3.3. Implementation of AMS Healthcare Facility Core Elements

Data gathered from the REDCap facilities assessment instrument showed significant improvements in AMS implementation stages for two core elements (Table 1). First, there were improvements in implementing regular ward rounds and other AMS interventions in selected facility departments conducted by the AMS team as identified in the AMS action plan (Z = 2.01, n = 25, *p* < 0.05). At baseline, 12% of facilities reported AMS team ward rounds and other interventions in the facility, 40% reported partial implementation, and 48% reported no implementation of this AMS action in the healthcare facility conducted by the AMS team. After one year, the proportion of facilities in the ‘fully implemented’ category increased to 32%, while the ‘not implemented’ category decreased to 24%. Second, there was a significant increase in the AMS team monitoring of antibiotic susceptibility and resistance rates for a range of key indicator bacteria in alignment with national and/or international surveillance systems (Z = 2.08, n = 25, *p* < 0.05). The reported data show baseline full implementation of this element by 32% of the facilities compared to 60% in the follow-up data.

Although not significant, the data suggest more facilities had dedicated, sustainable, and budgeted financial support for AMS activities in their AMS action plan after participating in the program (Z = 1.96, n = 25, *p* = 0.05). At baseline, 56% of facilities reported not having dedicated financial support for the AMS action plan, 36% reported partial support, and 8% reported full support. After one year, the proportion of facilities in the ‘fully implemented’ category doubled (from 8% to 16%), while the ‘not implemented’ category decreased to 36%. Additionally, the offering of initial and regular training of the AMS team in infection management and AMS increased from 32% at baseline to 60% in the follow-up assessment; however, three facilities reported offering the training at baseline and not offering it anymore after a year. Although the results do not show statistical significance, there is potential for the two trends above to be clinically meaningful, particularly given our small sample size. Table 1 shows results for all primary elements and secondary and tertiary nested elements, conditional on implementing one of the 14 primary elements. The last three columns summarize the percentage of facilities that obtained increased, unchanged, or decreased levels of implementation of a particular element when comparing the follow-up and baseline ratings. Increased or decreased levels may have included one or two steps, for instance, from no implementation to partial implementation, or from no implementation to full implementation. That level of granularity is not shown on the table but is considered in the statistical analysis. Despite the results outlined above, we observed limitations linked to using this online tool, which are discussed in later sections.

### 3.4. Practice and Systems Changes

Follow-up surveys were disseminated between February and March 2024, and we received 75 open-ended question responses in the Kenya survey, 25 in the Ghana survey, and 10 in the Malawi survey. All FGDs were conducted in May 2024, with 12 participants in total. The Kenya program FGD had three engaged participants: a medical laboratory technician, a pharmacist, and an AMS/Quality Improvement IPC project coordinator. Five participants joined the Ghana program FGD, including three physicians, a pharmacist, and a nurse. Finally, four participants shared their experiences in the Malawi program FGD, including a medical laboratory officer, a medical laboratory scientist, a clinical officer, and a nurse (the latter was also an IPC Coordinator and AMS Officer). Eight main themes emerged from the thematic coding of FGD transcripts and open-ended survey responses, as defined and detailed in Table 2.

#### 3.4.1. Prescription Practice, AMS Interventions, and Use of Microbiology Laboratory Themes

**Prescription Practice Improvements and AMS Interventions Applications** were the most common themes shared in participants’ FGDs. Both themes appeared in all three focus groups. The Prescription Practice Improvements highlights individual practices linked to learning from the TEACH AMS program, including using culture results to guide treatment and making evidence-based treatment choices, as illustrated in the quote below.


*“[During a TEACH AMS case presentation] the subject matter experts made us aware that [antimicrobial A] is useful for cystitis but not for pyelonephritis. And interestingly enough, about a week or two later, I had a patient with a similar presentation and (…) due to the information gathered from the session, there was enough confidence to immediately decide to go with [antimicrobial B].”*
(participant G03)

The AMS Interventions Applications theme refers to specific practices, such as participating in ward rounds and conducting point-prevalence surveys, or system-level processes, including creating, updating, or improving policies. For example, the quotes below, from a Kenya and a Ghana FGD participant, illustrate this theme:


*“We were able to do baseline assessments and specific tasks during the implementation. We were able to bring aspects of surveillance and diagnostic stewardship and dissemination of the findings through the AMS team and the health management team for actual policy change.”*
(participant K04)


*“The TEACH AMS platform has really impacted our practice in terms of antimicrobial prescribing in the hospital. (…) Any time a prescriber wants to prescribe a Watch or a Reserve antibiotic, a pharmacist has to be consulted before such an antibiotic is given. And per the policy we drafted from sessions from the TEACH AMS, cultures have to be requested (…) before such antibiotics are started.”*
(participant G07)

Similarly, combining follow-up survey responses, Prescription Practice Improvements, and AMS Interventions Applications were the most common themes shared by 45 and 38 out of 110 responses, respectively. The following quote from a Malawi participant illustrates the former: “*I took the time to only prescribe antibiotics if they were really needed according to lab results*” (respondent 092), and a Kenya participant quote exemplifies the later theme: “*We developed an AMS policy to guide implementation of AMS activities*” (respondent 005). Both themes were present in examples shared by all three countries’ survey respondents.

Another significant theme discussed in all three FGDs was **Improved Use of the Microbiology Laboratory** for proper case identification to inform therapy. Five participants discussed this theme, which includes examples of sending samples to the laboratory, confirming pathogen isolates, and creating local antibiograms, as exemplified in the quote below:

*“So one of the things that I have learned through these ECHO sessions is to draw samples first. (…) So with the prompt action on [maternal sepsis] to investigate and to take pus swabs for culture, and to take blood cultures at an earliest time, we have reduced some other cases that we usually had to refer to tertiary hospitals”*.(participant M06)

While this theme was not as common in follow-up survey responses as the previous themes discussed (23/110), it appeared in follow-up survey responses across the three countries. For example, a Ghana respondent wrote: “*Blood samples of suspected infections cases are taken before initiation of antibiotic therapy*” (respondent 073).

The three themes discussed above show strong evidence of the impact of the TEACH AMS programs in Kenya, Ghana, and Malawi. Still, it is important to note one particularity in the qualitative data disaggregated by program. While the Improved Prescription Practices and Application of AMS Interventions themes were evenly distributed in examples shared by participants across all programs, the Improved Use of Microbiology theme was more prominent for Malawi participants. Indeed, in follow-up surveys, this theme was the most common for Malawi respondents (7/10). Conversely, participants in the Ghana and Kenya FGDs mentioned improved use of the laboratory as a practice they changed because of participating in TEACH AMS, and examples were shared by 3/25 and 13/75 respondents in follow-up surveys, respectively.

#### 3.4.2. Other Qualitative Analysis Themes

This section outlines other qualitative themes that emerged in the analysis, from the most to the least prominent in the data. First, the **Communication Across Diverse Health Professionals** theme was present in all three FGDs. Five participants provided examples of improved communication, including information sharing between laboratory staff and clinicians, as exemplified below by a Kenya FGD participant:

*“We noted that different clinicians were not conversant with the different standard turnaround times for results (…). [Therefore] we had sensitization meetings from the lab, microbiology department in the different facilities. And clinicians were able to realize that the turnaround time for blood cultures was not the same as the turnaround for [other cultures]. (…) Most of the staff who joined the TEACH program from the facilities (…) [realized] there is need for us to dig deeper and understand patient care in collaboration with the other colleagues”*.(participant K05)

However, the follow-up survey response evidence for the Communication Across Diverse Health Professionals theme was not as strong as the themes discussed previously. A few survey respondents (12/110) shared examples of using a multidisciplinary approach or communicating across different health professionals as a practice that started due to TEACH AMS program participation, such as in the example below: “*having (…) lab and clinician interface which helps in sharing of ideas and also discussing on results and antibiotic use and the importance of lab requests and cascade reporting*” (respondent 021).

**Infection Prevention and Control Measures** also emerged as a theme, highlighting measures such as improved facility waste management programs and increased hand hygiene. The theme appeared in the Ghana and Kenya FGDs and in follow-up survey responses for program participants in those two countries. For example, a Ghana FGD participant shared the following: “*We have really involved the mothers, the patients, or the clients as well, as well as the staff when it comes to handwashing with segregation, cleanliness at the ward.*” (participant G06). Another example is this follow-up survey response from a Kenya participant: “*Implementing IPC and AMR/AMS programs has impacted the best practice in the facilities considered as sites. (…) Hand hygiene and surgical site infections surveillance is now practiced in our selected facilities and outcome is discussed*” (respondent 024). There was no evidence for practice or systems changes under this theme for Malawi participants.

The theme of **Education or Training of Healthcare Staff** only appeared in the Ghana FGD. Two participants made note of using knowledge gained from TEACH AMS to train their peers, as exemplified in this participant quote: “*At least within my department and with the younger doctors that I’m working with, (…), I make sure to hammer into them as nicely as I can that we should ensure that we take all the necessary cultures*” (participant G03). Ghana and Kenya survey respondents described experiences of educating and training other health professionals due to participating in the TEACH AMS programs. Quotes have informal examples, such as discouraging the use of unnecessary antimicrobials or advocating for the implementation of AMS strategies, and formal examples, including new topics in training. This theme was not shared by any of the Malawi survey respondents.

The themes of **Improved Communication with Patients and/or Community and Advances in the Facility AMS Committee** were shared as examples by fewer than five survey respondents when combining the three programs’ survey responses, indicating they were not as significant for participants as were other practices they shared as examples of applications from session learnings. Nevertheless, they indicate potential changes with regard to improving facility AMS Committees and effectively communicating with patients and communities about AMS. For example, the respondent of the Malawi survey wrote the following: “*I took the time to (…) [discuss] with the patients and guardians on why the antibiotics are not really necessary in their cases so they understand why I’m not prescribing and not get them elsewhere*” (respondent 092), and a Kenya survey respondent shared the following: “*I planned community dialogue to sensitize community members of the risk of developing resistance on the abused drug*” (respondent 046). The Improved Communication with Patients and/or Community theme was discussed by Ghana FGD participants only. The quote below focuses on bringing the family of pediatric patients into the care plan and educating them to prevent and lessen infections:


*“(…) if a child is diagnosed with UTI, for instance, the mother is taught to constantly change the diapers, wash their hand (…). We educate [parents] very well as to some of the things that they are supposed to do so that the hospital stay will be reduced”*
(participant G06)

For the Advances in the Facility AMS Committee theme, there were two examples shared by Kenya survey respondents, one about establishing a committee at the facility and the other referring to having more frequent meetings to help communicate across different professionals. Similar to the previous theme, the Advances in the Facility AMS Committee theme was discussed only in the Kenya FGD by two participants who shared insights about how participation in TEACH AMS sessions may have motivated local AMS Committee teams, as exemplified in the quote below:


*“Following these TEACH AMS sessions, we were able to regroup and refocus [in the Antimicrobial Stewardship Committee]. (…) We were also able to bring in the administration and have a budget allocated for AMS activities”*
(participant K04)

#### 3.4.3. Barriers to Knowledge Application

In follow-up surveys, respondents were prompted to select barriers to applying what they learned in the TEACH AMS sessions in their work (all that applied). We received 252 responses to the question about barriers, 143 from Kenya, 74 from Ghana, and 35 from Malawi program participants. Overall, the most commonly selected barriers across all respondents were the lack of resources and the need for more training (Figure 3). For Malawi, over 30% of respondents also selected the lack of supporting guidelines as a barrier.

In all FGDs, participants identified a lack of resources as the main barrier to applying best practices learned in the TEACH AMS programs, corroborating the follow-up survey evidence. In the Malawi and Kenya FGDs, participants shared that appropriate antimicrobials are not always available at their facilities, leading to dire consequences for patients. Similarly, in the Ghana and Malawi FGDs, participants elaborated by saying that patients face financial constraints and often cannot afford to purchase the required prescribed antimicrobials:

*“(…) Most of the time, the antibiotics that matter, that we need to dispense to the patients, are not available. (…) even if you encourage the patients to buy, they will tell you, ‘I don’t have funds to buy such a drug’”*.(participant M06)

Participants in the Kenya and Ghana FGDs also discussed a lack of laboratory materials and capacity to conduct specific microbiology tests. In the Kenya FGD, participants mentioned high workload barriers and behavioral and systems challenges with the timely review and documentation of the patient’s history in health records systems. In the Ghana FGD, participants alluded to administrative or leadership support issues related to intractable and long decision-making processes for obtaining resources at the facility and, nationally, the need for improved regulatory systems. FGD participants did not mention training barriers.

## 4. Discussion

Our findings indicate that a virtual telementoring model can support AMS capacity building across diverse health care settings in LMICs. The TEACH AMS ECHO programs in Kenya, Ghana, and Malawi were feasible and far-reaching in building capacity among multi-disciplinary healthcare providers to implement AMS interventions in targeted hospitals, as reflected in program attendance, participant satisfaction, knowledge gains, and reported systems and practice level changes. The extensive reach of the programs, with 77 learning sessions which engaged over 2400 unique participants, feedback indicating high rates of self-reported knowledge gain and satisfaction with the relevance of the sessions, and the fact that more than 60% of attendees returned to participate in additional learning sessions, builds on the evidence of previous studies about the success of virtual learning approaches for healthcare providers to address AMR [21,22,24]. Participation from countries outside the targeted geographies also highlights the broad interest and acceptability of virtual AMS training opportunities across LMIC in Africa. The results of the TEACH AMS evaluation suggest that the virtual case-based learning ECHO model is not only feasible in sub-Saharan Africa but is an acceptable and successful tool for engaging multi-disciplinary health care workers and building AMS capacity. To our knowledge, this is the first study to document the impact of a virtual learning approach on provider practice and systems changes and increases in facility AMS capacity in LMICs.

Additionally, after approximately 17 months of on-going TEACH AMS ECHO educational sessions in the three countries, there were significant facility level improvements, particularly in two key AMS core elements: implementing regular ward rounds and other AMS interventions in selected facility departments conducted by the AMS team; monitoring of antibiotic susceptibility and resistance rates for a range of key indicator bacteria. These represent foundational changes critical for sustained AMS success. These changes were further substantiated by findings from follow-up surveys and FGDs. The focus on behavior change has led to improvements in prescribing practices and the application of AMS interventions, such as conducting regular ward rounds, as outlined in the AMS action plans of healthcare facilities. However, these improvements were self-reported, and objective outcome measures would be desirable to confirm such impacts in future studies.

Differences in context explain the variability in outcomes across the TEACH AMS ECHO programs. For example, Kenya is a larger and more populous country, and its program was built upon a well-established and long-standing laboratory AMR ECHO, attracting more pharmacists and achieving greater reach compared to the programs in Ghana and Malawi. Our study data also indicate that certain best practices were more commonly applied in specific programs. For instance, participants from Malawi emphasized improvements in the use of microbiology laboratories as a result of their program learning. In contrast, participants from Kenya and Ghana highlighted the importance of communication among diverse health professionals and the implementation of specific IPC practices. Despite these differences, it is imperative to highlight that participants in all three programs reported applying similar core practices and implementing essential elements of AMS programs within healthcare facilities.

The TEACH AMS programs in each country attracted participants from diverse disciplines, as demonstrated by the attendance data. The initiative’s reach highlights the opportunity the virtual TEACH AMS ECHO sessions offered to foster multi-disciplinary engagement, leveraging various expertise and perspectives, and creating communication channels critical for implementing AMS strategies. The case-based learning approach further catalyzed inter-disciplinary communication at the facility level, as reported by participants in follow-up surveys and FGDs, and observed in program design, as the different professions frequently come together to prepare their cases for the session. The virtual nature of the TEACH AMS ECHO sessions, as well as local coordination from the MOH AMR focal points and engagement with professional societies, and the provision of Continuing Professional Development (CPD) points for multiple cadres in Kenya and Ghana, were successful approaches to encourage multidisciplinary participation in the TEACH AMS ECHO sessions.

While data are limited for LMICs, previous studies have found that interventions that combine multiple approaches, such as education combined with interventions like prescription audits and policy restrictions, are more effective than single-faceted interventions [46,47,48] in changing antimicrobial prescribing practices of providers, though education as a standalone intervention has been shown to have a positive impact [47]. The TEACH AMS programs, while centered on education, also brought together key stakeholders and created opportunities for bi-weekly engagements that focused on real-world issues and challenges (“cases”) presented by a diverse group of learners. This ensured that the problems, discussions, and solutions in the TEACH AMS ECHO sessions were informed and driven by the local context and relevant to the health care workers. The literature confirms the success of this approach as local partnerships and contextualization are critical in developing AMR and AMS interventions and tools, and sustained behavior change is best achieved when AMR and AMS training are tailored and ongoing [17].

The TEACH AMS initiative occurred concomitant to other local AMR/AMS activities for laboratory strengthening, data collection, and other facility-based AMR/AMS training modules. However, the data collected from the facility assessment, combined with the high satisfaction and knowledge gain reported in post-session surveys and the specific examples of practice and systems changes shared by TEACH AMS participants in follow-up surveys and FGDs, underscore the TEACH AMS programs’ impact on increased capacity and actual implementation of practice and systems changes to address AMR. Although increases in AMS team training did not reach statistical significance, the observed trend (from 32% to 60%) suggests positive momentum. These findings align with WHO guidance emphasizing structural readiness and workforce education as prerequisites for AMS program maturation. Additionally, there are substantial cost savings achieved when using virtual trainings instead of in-person trainings [27], which should be considered when developing educational approaches.

Some of the barriers reported by participants in implementing the knowledge gained from the TEACH AMS sessions included a lack of resources, such as the availability of antimicrobials and laboratory supplies. Notably, the need for further training was also a frequently cited barrier in the surveys, though not raised in the FGDs. This may be due to a selection bias, as FGD participants are more likely to be those who are more engaged and have received advanced training. Systemic limitations, such as the need for training and unavailable critical resources, often hinder the ability to act on best practices, even when awareness and intent are high. Given the local implementation of the TEACH AMS ECHO approach, MOH partners can continue to evolve and adapt the TEACH AMS ECHO sessions to meet ongoing and emerging training needs, and use them as opportunities to identify knowledge and resource gaps. The increased awareness among providers about AMS interventions and the opportunity for ongoing engagement with the MOH may also contribute to increased advocacy and assist with targeting resources to the most critical areas to better support AMS implementation. Overcoming these barriers will require not only educational interventions but also policy-level commitment and sustainable investment in healthcare infrastructure.

Barriers to implementing the TEACH AMS ECHO approach included connectivity/internet challenges and the lack of in-country SMEs for some topics. While many facilities do have internet infrastructure, the provision of data bundles for some facilities was needed in order to ensure they could regularly connect. To address the need for subject matter expertise, a standard core TEACH AMS curriculum developed in partnership with the American Society of Microbiology and in-country SMEs provided content that the programs could use and adapt as needed. In addition, the virtual nature of the ECHO learning approach facilitates participation of external SMEs, and relationships established between the countries as part of the overall TEACH AMS collaborative meetings created opportunities for regional expertise to be leveraged across the different TEACH AMS programs.

One of the main limitations of our study includes low survey response rates. The 8.2% response rate in post-session surveys, in particular, limits the generalization of our program satisfaction and knowledge-gain findings. The post-session survey implementation also posed challenges since busy health care workers lacked the time to respond to the survey after every session, and session organizers struggled to cover content and limit case discussions within the timeframe, and rarely provided time at the end of sessions for participants to complete the survey as recommended. Additionally, relying on self-reported data may introduce bias. For instance, social desirability bias may have been present, especially in open-ended questions or FGDs. Moreover, while the online REDCap tool was useful for rapid data collection and management, it presented challenges in ensuring consistent and accurate facility-level reporting. Ideally, the assessment data could have been cross-checked with on-the-ground objective observations or data, for example, by collecting antibiotic consumption data from facilities. These types of data collection were not implemented as they were beyond the scope of our initiative and would require additional resources. Furthermore, the short time frame between program initiation and follow-up assessment may not fully capture the long-term sustainability of the observed changes. Lastly, evaluations of virtual training are often found in less formal sources, such as online reports or conference abstracts, which are not available in the databases we researched. This makes it more difficult to compare our results with other studies.

Despite the limitations outlined above, our study has several strengths. For example, while the post-session survey response rate was low, the significant knowledge gain (*p* < 0.0001) adds rigor to the findings. This suggests that even limited but targeted engagement can result in meaningful learning, though strategies to increase response rates should be considered in future implementation. Also, the adaptation of the WHO Facility AMS Capacity assessment tool into an online instrument to collect baseline and follow-up AMS capacity data represents an important innovation of this study. While previous and more extensive deployments of this tool have occurred and have assisted with identifying AMS capacity gaps to inform and guide country AMS implementation [16,17], the adaptation and use of the tool in combination with qualitative evaluation methods to assess the impact of a virtual AMS capacity building initiative is a new approach. The TEACH AMS initiative leveraged virtual education to reach a wide and diverse audience with relatively low resource input. The use of a mixed-methods approach, combining quantitative tracking, surveys, and qualitative FGDs, strengthened the validity and depth of the findings.

Our findings from the TEACH AMS case-based learning programs in Kenya, Ghana, and Malawi focused specifically on one component of the multifaceted approaches needed to successfully implement AMS programs, highlighting the potential effectiveness of virtual communities of practice as a complementary educational intervention. These communities, formed by key stakeholders, experts, and direct health providers, have a substantial and multidisciplinary reach, indicating that they play a critical role in driving behavior change, such as promoting evidence-based prescribing and treatments and improving AMS processes and policies tailored to local contexts for optimal patient care and outcomes. Future studies could further strengthen the evidence for the success of such educational methods in AMS by using case-control designs, incorporating objective facility data such as intervention costs and percentage of antimicrobial prescriptions by category as monitored by health facilities. Additionally, capacity-building efforts should focus on sustaining and scaling virtual telementoring initiatives. This will help address local challenges, enhance effective communication among diverse professionals, and strengthen AMS activities within health facilities.

## Figures and Tables

**Figure 1 antibiotics-14-00794-f001:**
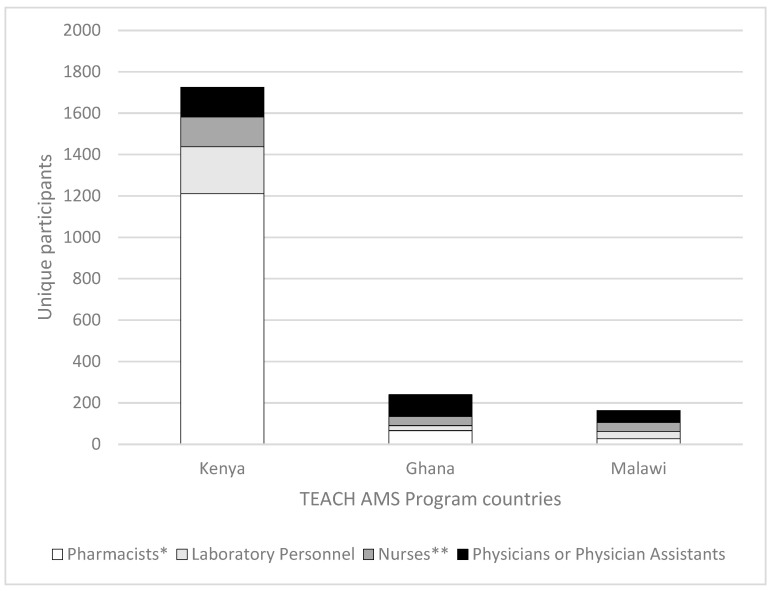
Distribution of the most common professions of participants in the Kenya, Ghana, and Malawi programs. * The “Pharmacists” category includes pharmacy technicians and pharmacy faculty. ** The “Nurses” category includes many nurse specialties and nurse faculty (N = 2128 for all programs, 1725 for Kenya, 240 for Ghana, and 163 for Malawi).

**Figure 2 antibiotics-14-00794-f002:**
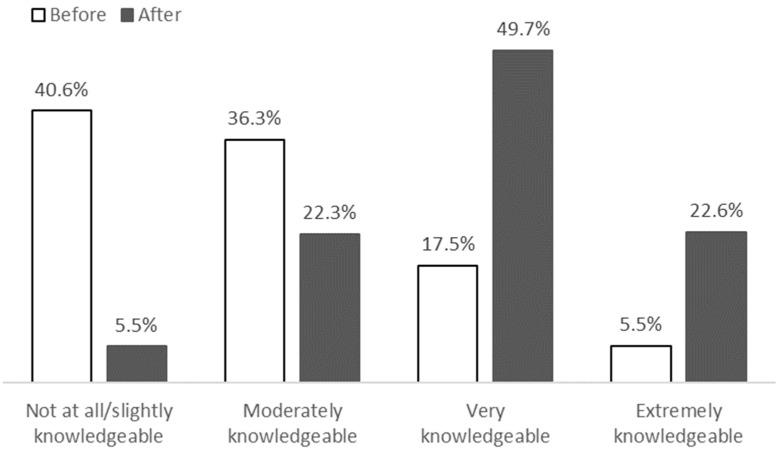
Before and after subjective session topic knowledge post-session survey ratings (N = 725).

**Figure 3 antibiotics-14-00794-f003:**
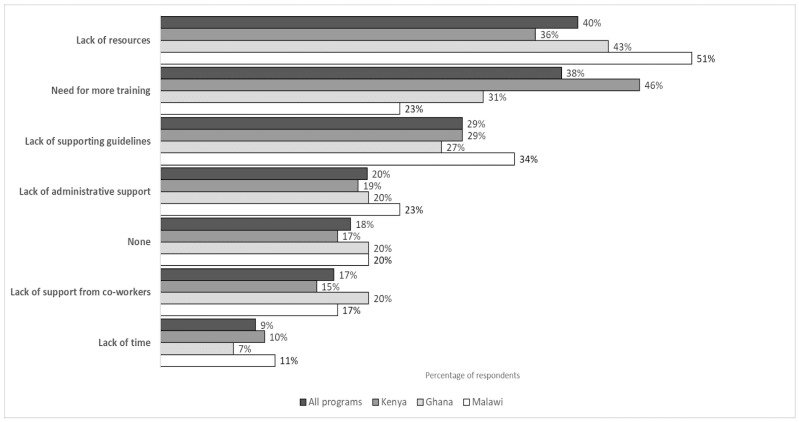
Barriers to knowledge application reported by follow-up survey respondents (N = 252).

**Table 1 antibiotics-14-00794-t001:** Baseline and follow-up assessment comparison results of implementation levels of healthcare facility core elements for AMS programs in LMICs (N = 25).

**Core Primary AMS Components**	**Z**	**p-Value**	**Increased**	**Unchanged**	**Decreased**
AMS prioritized with management action plan	1.9	0.06	32%	60%	8%
Accessible laboratory and imaging services	1.7	0.08	12%	88%	0%
Antibiogram informed by antimicrobial use and resistance data	1.6	0.11	36%	52%	12%
Dedicated AMS leader/champion identified	1.4	0.17	16%	80%	4%
Multidisciplinary AMS leadership committee with clear ToR	1.3	0.18	16%	80%	4%
Standardized facility prescription charts and medical records	1.4	0.2	16%	80%	4%
Antimicrobial use regularly evaluated and shared	−1.1	0.3	8%	72%	20%
Basic training in optimal antimicrobial use	0.9	0.4	32%	48%	20%
Resistance rates regularly evaluated and shared	0.9	0.4	32%	48%	20%
Up-to-date standard treatment guidelines	0.7	0.5	20%	68%	12%
Accessible IT services to support AMS activities	0.6	0.5	24%	60%	16%
Approved antimicrobials list	0.5	0.6	24%	56%	20%
Policy for documenting prescribed medicines	0.5	0.6	20%	68%	12%
Restricted antimicrobial list and implementation guidelines	0.4	0.7	24%	60%	16%
**Core Secondary AMS components**	**Z**	**p-value**	**Increased**	**Unchanged**	**Decreased**
Dedicated financial support for AMS action plan	2	0.05	48%	36%	16%
Multidisciplinary AMS team with ToR	1.2	0.22	44%	44%	12%
AMS action plan endorsed with progress and accountability measures	1.2	0.23	44%	40%	16%
Continued training in optimal antimicrobial use	0.82	0.41	40%	40%	20%
**Core Tertiary AMS components**	**Z**	**p-value**	**Increased**	**Unchanged**	**Decreased**
Regular ward rounds and other interventions by AMS team in select departments	2.0	0.04 (*)	48%	36%	16%
Monitoring antimicrobial susceptibility and resistance rates for a key indicator bacteria	2.1	0.04 (*)	44%	44%	12%
Initial and regular training of the AMS team in infection management	1.8	0.06	44%	40%	16%
Monitoring of compliance of AMS interventions by AMS committee	1.4	0.16	40%	40%	20%
Monitoring of quantity and types of antimicrobial use (purchased/prescribed/dispensed)	1.4	0.17	32%	56%	12%
Defined collaboration between the AMS and IPC	1.3	0.18	40%	36%	24%
Regular (descriptive) activity reports on AMS implementation	1.1	0.25	36%	44%	20%
Regular activity reports (status and outcomes) on AMS implementation	1.1	0.25	36%	44%	20%
Regular AMS team review/audit (antimicrobial therapy or clinical conditions)	1.1	0.27	40%	36%	24%
Audits or PPSs monitoring for appropriate antimicrobial use	1.0	0.31	36%	44%	20%
AMS team feedback easily available to all prescribers	1.0	0.33	40%	36%	24%
Other health professionals identified and involved in AMS activities	0	1	12%	64%	24%

* The results are statistically significant at the 0.05 level.

**Table 2 antibiotics-14-00794-t002:** Focus group discussion and follow-up survey open question codebook.

Themes	Inclusion Criteria/Definition
Prescription Practice Improvements	Quotes from focus group participants or examples shared in follow-up surveys open questions in which respondents mentioned or wrote about improved antimicrobial prescription at the individual level, including use of culture results for prescribing, prescribing antimicrobials supported by laboratory results, use of target therapy, ensuring right frequency and dose is prescribed, and others, after participating in the TEACH AMS program.
AMS Interventions Application	Quotes from focus group participants or examples shared in follow-up surveys open questions in which respondents mentioned or wrote about implementing AMS activities, including use of the AWaRe classification (Access, Watch, reserve WHO system), conducting prescription audits, conducting ward rounds and panels, implementing point-prevalence survey, antibiotic inventory management, review of antibiotics after 48 h, creating or using antibiotics guidelines, SOPs, and workplans, creating restricted antimicrobials policy, cost implications, diagnostic stewardship, policies regarding any of the previously listed AMS interventions, system levels improvements of prescriptions, and others, after participating in the TEACH AMS program.
Improved Use of Microbiology Laboratory	Quotes from focus group participants or examples shared in follow-up surveys open questions in which respondents mentioned or wrote about improved sample collection and processing; improvements in test quality, safety, and efficiency; regular control testing for facility antibiotics for AST; confirming isolates; and having cultures done before prescribing antimicrobials, after participating in the TEACH AMS program.
Education or Training of Health Care Staff	Quotes from focus group participants or examples shared in follow-up surveys open questions in which respondents mentioned or wrote about educating other health care workers (e.g., discouraging the use of unnecessary antimicrobials, including specific learnings in facility trainings, presentations at the facility, advocating for proper and correct use of antimicrobials in facilities, advocate for forming AMS committee, advising prescribers) about AMS topics after participating in the TEACH AMS program.
IPC measures	Quotes from focus group participants or examples shared in follow-up surveys open questions in which respondents mentioned or wrote about infection, prevention, and control or facility waste management, often accompanied by examples such as equipment decontamination, hand washing, restricted entry to possibly contaminated areas, WASH, and others after participating in the TEACH AMS program.
Communication Across Diverse Health Professionals	Quotes from focus group participants or examples shared in follow-up surveys open questions in which respondents mentioned or wrote about applying a multidisciplinary approach, membership, or similar in their AMS-related activities in the workplace, communicating across diverse professional staff, after participating in the TEACH AMS program.
Communication with Patients and/or Community	Quotes from focus group participants or examples shared in follow-up surveys open questions in which respondents mentioned or wrote about educating patients, guardians, and/or the community after participating in the TEACH AMS program.
Advances in the Facility AMS Committee	Quotes from focus group participants or examples shared in follow-up surveys open questions in which respondents specifically mentioned or wrote about AMS committee improvements, including reactivating or participating in the committee, after participating in the TEACH AMS program.

## Data Availability

The raw data supporting the conclusions of this article can be made available by the authors based on a reasonable request and according to ethics guidelines.

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
