# Peer review of "Bridging the Capacity Building Gap for Antimicrobial Stewardship Implementation: Evidence from Virtual Communities of Practice in Kenya, Ghana, and Malawi"

_antibiotics, 2025, doi:10.3390/antibiotics14080794_

Round 1
Reviewer 1 Report
Comments and Suggestions for Authors
1. In Europe more adequate is "Problem Based Learning" rather than ‘problem-posing education' - leaving for authors decission.
2. The introduction is definitely too long. It would be good to provide some baseline data defining the level of AMS and justifying the intervention
3. Results Table 1 will be more benficial and informative to show percetages prior and post intervention.
4. Do You have data about AMS team members? It will be beneficial to know who was taking part in wards (phisicians, pharmacologists, nurses)
5. Do you have more financial data? it will be better to show from which to which level was financing shifted, it gives information to readers how those interventions are cost consumming
6. It would be better to meassure AMS implementation someway for example perecent of prescription i AWARe groups oraz wide/brode spectrum rate to get hard MAS intervention results. In prospective study it is not so hard to plan this action.
7. According to financial problems it would be good to write about financing models in participating countries
The study is a good idea, but on the planning stage there is lack of planned hard data to support for example gaining knowledge, AMS practices etc. However after some improvements it may be basis for further studies.
Author Response
Comment 1: In Europe more adequate is "Problem Based Learning" rather than ‘problem-posing education' - leaving for authors decission.
Response 1: We discussed this issue internally and decided to maintain the language since it is the same as most of the literature cited in the article.
Comment 2: The introduction is definitely too long. It would be good to provide some baseline data defining the level of AMS and justifying the intervention.
Response 2: We shortened the introduction and provided background information on AMS implementation and gaps in the three countries. We also included references from the published literature regarding the importance of AMS to address AMR, and the current status and gaps in AMS implementation in LMIC
Comment 3: Results Table 1 will be more benficial and informative to show percetages prior and post intervention.
Response 3: We included the percentages as recommended, and included additional explanations to help with the interpretation.
Comment 4: Do You have data about AMS team members? It will be beneficial to know who was taking part in wards (phisicians, pharmacologists, nurses) .
Do you have more financial data? it will be better to show from which to which level was financing shifted, it gives information to readers how those interventions are cost consuming.
Response 4: We appreciate the thoughtfulness of the reviewer's comments. With regards to information on the AMS teams and participants, we have included information noting the leadership of the TEACH AMS programs with the MOH and academic partners, as well as professional distribution of the participants. While we agree that having the specific types of data listed would make for a more robust study including subjective and objective outcomes, objective on-the ground data collection such as who was taking part in wards or percent of prescription of certain antimicrobials and details on different financing models and cost-benefit analysis were impractical and outside the scope of our multi-country initiative's evaluation. A priority for the initiative was that the three programs described were context-specific, far-reaching, voluntary, and open to all. We opted to employ a mixed-methods approach, using diverse self-assessment data collection tools as the best design in this case, which aligned with the evaluation scope. That said, we have revised the language in the limitations section to call attention to those issues. We also revised the language in the design section, and reframed part of the discussion to focus on feasibility and recommend the use of objective measures with examples in future studies.
Reviewer 2 Report
Comments and Suggestions for Authors
The manuscript represents a mixed methods approach study conducted across three countries in Africa. The study is extensively conducted and the methods and results presented in a comprehensive manner. Few queries are:
Some details on study design should be added. For example,
- Pre/post knowledge assessment: was a questionnaire used for this? If yes, details on the same such as how and by whom was the questionnaire designed, was it validated before administration; domains asked, scoring etc.
- How many FGDs were conducted in total?
- The details on qualitative data analysis are to be added. The themes and sub-themes generated from qualitative data can be represented in a table for better understanding.
Author Response
Comment 1: Pre/post knowledge assessment: was a questionnaire used for this? If yes, details on the same such as how and by whom was the questionnaire designed, was it validated before administration; domains asked, scoring etc.
Response 1: We thank the reviewer for pointing out the need to detail and clarify some aspects of the study design. We added a paragraph with a more detailed description of the design, justification, and survey tool used for the retrospective pre- and post-knowledge assessment. The survey has not been formally validated; however, it has been used extensively across many international ECHO programs, primarily to provide rapid feedback and facilitate program improvement. The survey also includes questions about program satisfaction and inquiries into how the program team can address participants' needs.
Comment 2: How many FGDs were conducted in total?
Response 2: To provide more straightforward information about the study design and tools, we changed the language to make it clearer that we conducted three FGDs, one per country.
Comment 3: The details on qualitative data analysis are to be added. The themes and sub-themes generated from qualitative data can be represented in a table for better understanding.
Response 3: Table A2 in the Appendix outlined the themes derived from the qualitative data. Given how critical those definitions are to more easily understanding the results section, we moved it to the main part of the manuscript.
Reviewer 3 Report
Comments and Suggestions for Authors
This manuscript reports on the implementation of a virtual antimicrobial stewardship (AMS) training program using the ECHO model across healthcare institutions in Kenya, Ghana, and Malawi. The topic is timely and highly relevant, particularly in the context of capacity building for AMS in low- and middle-income countries . The use of mixed methods and a multi-country design strengthens the scope of the work.
However, the manuscript has methodological, and structural weaknesses that must be addressed before it can be considered for publication. The following changes may contribute to a better understanding to the readers o ft this journal.
Abstract
The results exposed may overstates the program’s effectiveness without acknowledging the very low survey response rate (8.2%) or design limitations. Rephrase the conclusion more cautiously (e.g., “suggests potential for impact” rather than “demonstrates effectiveness”).
Introduction
The introduction section may be too long and redundant in some aspects, especially in the last paragraphs. The informatiion related with ECHO model could be shortened
Methods
The methodology has serious limitations that must be corrected or explained:
- No objective clinical outcomes measured (e.g., antibiotic use, compliance with guidelines, resistance rates).
- Most measures are self-reported, prone to recall and social desirability bias.
- Qualitative data from only three focus groups (12 participants total), not representative of the entire cohort.
Justify the use of retrospective self-assessment surveys and describe any measures to mitigate bias. Include in tha analysis why no objective outcome measures were used.
Results
Recommendation:
- Figure 3 on barriers reports N=252. However, the text describing the follow-up surveys mentions 110 total responses to open-ended questions (75 from Kenya, 25 from Ghana, and 10 from Malawi). Please clarify this discrepancy
- With a small sample of N=25 facilities, the study may be underpowered to detect statistically significant changes. While only two elements show significance , others show positive trends (e.g., dedicated financial support, p=0.05; initial AMS team training, p=0.06). The authors could cautiously note these trends as potentially clinically meaningful, while clearly stating they did not reach statistical significance
Discussion
The discussion lacks critical analysis and includes overstatements:
- Describes the intervention as “effective” despite minimal statistically significant results and no objective clinical outcome data. Reformulate the discussion with a focus on feasibility and potential, not effectiveness.
- Comparisons to prior studies (e.g., other ECHO or AMS programs) are superficial and do not explore contextual differences.
- The post-session survey response rate of 8.2% is extremely low and presents a high risk of response bias. The respondents are likely not representative of the entire participant cohort. This methodological weakness must be more critically addressed, as it severely limits the generalizability of the satisfaction and knowledge-gain findings.
- Include a brief discussions of barriers to implementation (limited internet access, time constraints, institutional support, etc).
- The discussion should address the interesting discrepancy where "need for more training" was a top barrier in surveys but was not mentioned in FGDs. Speculating on the reasons for this (e.g., FGD participants being more engaged "super-users") would add valuable nuance to the paper
- The total number of FGD participants is very small (N=12) for a program with over 2,400 participants. The authors should be more cautious with the conclusions drawn from this data. Furthermore, the method for selecting these 12 individuals from the pool of those who "expressed interest" should be clarified to address potential bias.
- Address the likely Hawthorne effect and selection bias in those who responded to surveys or engaged most actively.
Limitations
- Some relevant limitations are listed. However. The absence of baseline data for most outcomes, lack of control group, and overreliance on self-assessment are not sufficiently emphasized. Explicitly note that antibiotic consumption, was not assessed.
Conclusions
- Replace statements like “this study demonstrates…” with “this study suggests…”.
- Emphasize the value of the training model as a potentially scalable framework, but caution that further research is needed to validate impact.
In summary, this manuscript has the potential to be a valuable contribution to the literature. However, it is currently undermined by a critical error in its timeline and methodological weaknesses that are not adequately addressed. A major revision is required to correct these issues and present a more balanced and credible interpretation of the findings.
Author Response
Comment 1: The results exposed may overstates the program's effectiveness without acknowledging the very low survey response rate (8.2%) or design limitations. Rephrase the conclusion more cautiously (e.g., "suggests potential for impact" rather than "demonstrates effectiveness").
Response 1: We agree with the reviewer and have rephrased the conclusion section of the abstract as recommended.
Comment 2: The introduction section may be too long and redundant in some aspects, especially in the last paragraphs. The informatiion related with ECHO model could be shortened
Response 2: We agree with the reviewer and shortened the introduction section, particularly the paragraph about case-based learning and the ECHO model.
Comment 3: The methodology has serious limitations that must be corrected or explained:
- No objective clinical outcomes measured (e.g., antibiotic use, compliance with guidelines, resistance rates).
- Most measures are self-reported, prone to recall and social desirability bias.
- Qualitative data from only three focus groups (12 participants total), not representative of the entire cohort.
Response 3: While we agree with the reviewer that it is desirable to document and evaluate programs using objective clinical outcomes, this was not the scope of the program evaluation. We opted to employ a mixed-methods approach, using diverse data collection tools (e.g., registration, post-session surveys, follow-up surveys, and FGD), which aligned with the evaluation scope. Moreover, collecting standard clinical outcomes data across organizations and countries would require financial and logistical resources beyond our study budget. Given the above, we don’t see the lack of objective clinical outcomes data as a serious limitation. Moreover, methodologically, focus group discussions are often not representative of the population studied. Nevertheless, they provide critical and valid evidence of participants' experience, as well as facilitators and barriers to practice changes. In the text, we emphasized that we did not collect clinical outcomes. We added the following language in the study design section: "We did not collect clinical outcomes data, as it was outside the scope of this study and would have required logistical and financial resources that were unavailable to the study team."
Comment 4: Justify the use of retrospective self-assessment surveys and describe any measures to mitigate bias. Include in that analysis why no objective outcome measures were used
Response 4: We added a paragraph with a more detailed description of the design, justification with a citation, and survey tool used for the retrospective pre- and post-knowledge assessment. Response 3 addresses the comment about the lack of objective measures used.
Comment 5: Figure 3 on barriers reports N=252. However, the text describing the follow-up surveys mentions 110 total responses to open-ended questions (75 from Kenya, 25 from Ghana, and 10 from Malawi). Please clarify this discrepancy
Response 5: The manuscript was unclear regarding the description of the follow-up survey. We revised the language in the design section and included the multiple-choice 'N' for follow-up surveys in each country in the Results section.
Comment 6: With a small sample of N=25 facilities, the study may be underpowered to detect statistically significant changes. While only two elements show significance , others show positive trends (e.g., dedicated financial support, p=0.05; initial AMS team training, p=0.06). The authors could cautiously note these trends as potentially clinically meaningful, while clearly stating they did not reach statistical significance
Response 6: We agree with the reviewer that the facility assessment data sample size is a limitation of our study. We revised the language in the manuscript as suggested by the reviewer.
Comment 7: The discussion lacks critical analysis and includes overstatements:
- Describes the intervention as “effective” despite minimal statistically significant results and no objective clinical outcome data. Reformulate the discussion with a focus on feasibility and potential, not effectiveness.
- Comparisons to prior studies (e.g., other ECHO or AMS programs) are superficial and do not explore contextual differences.
- The post-session survey response rate of 8.2% is extremely low and presents a high risk of response bias. The respondents are likely not representative of the entire participant cohort. This methodological weakness must be more critically addressed, as it severely limits the generalizability of the satisfaction and knowledge-gain findings.
- Include a brief discussions of barriers to implementation (limited internet access, time constraints, institutional support, etc).
- The discussion should address the interesting discrepancy where "need for more training" was a top barrier in surveys but was not mentioned in FGDs. Speculating on the reasons for this (e.g., FGD participants being more engaged "super-users") would add valuable nuance to the paper
- The total number of FGD participants is very small (N=12) for a program with over 2,400 participants. The authors should be more cautious with the conclusions drawn from this data. Furthermore, the method for selecting these 12 individuals from the pool of those who "expressed interest" should be clarified to address potential bias.
- Address the likely Hawthorne effect and selection bias in those who responded to surveys or engaged most actively.
Response 7: We agree with many of the reviewers' points regarding the discussion and appreciate the recommendation to reframe it with a focus on feasibility. We highlighted the potential bias linked to the low response rate of surveys, and discussed the use of three FGD, including the Hawthorne effect and selection bias. We also addressed the discrepancy between the follow-up survey and FGD data, adding nuance. The relatively small number of publications on virtual education and training approaches for AMR/AMS education, most of which did not employ the ECHO case-based learning approach, limited our ability for comparison. Instead we have summarized the overall approaches and the outcomes of other studies on virtual learning approaches to note the feasibility of virtual education and training, while highlighting some of the differences in the TEACH AMS ECHO approach and some of the gaps that we feel our manuscript helps to address.
Comment 8: Some relevant limitations are listed. However. The absence of baseline data for most outcomes, lack of control group, and overreliance on self-assessment are not sufficiently emphasized. Explicitly note that antibiotic consumption, was not assessed.
Response 8: We have revised the language in the limitations section and included the reviewer's recommendations. While we agree with the reviewer that causality studies often require case-control and objective measures, these types of data collection and designs were impractical and outside the scope of our multi-country initiative's evaluation. A priority for the initiative was that the three programs described were context-specific, far-reaching, voluntary, and open to all. We opted to employ a mixed-methods approach, using diverse self-assessment data collection tools as the best design in this case, which aligned with the evaluation scope. Moreover, the design suggested by the reviewer would also require financial and logistical resources beyond our study budget.
Comment 9: Replace statements like “this study demonstrates…” with “this study suggests…”.
- Emphasize the value of the training model as a potentially scalable framework, but caution that further research is needed to validate impact.
Response 9: We agree with the reviewer's assessment and have changed the language accordingly.
Round 2
Reviewer 1 Report
Comments and Suggestions for Authors
I thank the authors for considering the submitted comments. I understand the limitations of funding. Unfortunately, many such programs and interventions in the field of AMS or infection control are associated with low feedback rates, making evaluation of the interventions difficult. Future studies should develop models that encourage feedback and plan the use of more robust clinical data. I encourage the authors to continue their work given the importance of this issue.
Reviewer 3 Report
Comments and Suggestions for Authors
The revised manuscript successfully addresses all the suggestions and points made. Congratulations to the authors for their work.